# Tear and Plasma Levels of Cytokines in Patients with Uveitis: Search for Active Disease Biomarkers

**DOI:** 10.3390/jcm11237034

**Published:** 2022-11-28

**Authors:** Luis García-Onrubia, Milagros Mateos Olivares, Carmen García-Vázquez, Amalia Enríquez-de-Salamanca, Lidia Cocho, José María Herreras Cantalapiedra

**Affiliations:** 1Department of Ophthalmology, Hospital Clínico Universitario de Valladolid, Avda Ramón y Cajal 3, 47003 Valladolid, Spain; 2IOBA (Institute of Applied Ophthalmo Biology), Universidad de Valladolid, Paseo de Belén 17, 47011 Valladolid, Spain; 3Biomedical Research Networking Center in Bioengineering, Biomaterials and Nanomedicine (CIBER-BBN), 28029 Madrid, Spain

**Keywords:** uveitis, biomarkers, intraocular inflammation, tears

## Abstract

Uveitis accounts for up to 20% of blindness in Europe, making the development of new non-invasive biomarkers which could help in its management a field of interest. It has been hypothesised that tear levels of cytokines and chemokines could be used as a potential biomarker in patients with anterior uveitis, and this could be correlated with their concentration in plasma. Therefore, we measured twelve cytokines/chemokines in tear and plasma samples of 22 patients diagnosed with active anterior uveitis. Levels of these molecules in tears and plasma were compared and associated with the degree of activity of the uveitis. It is notable that the percentage of tear interleukin (IL)-6 detection was significantly reduced in the inactive phase (*p* < 0.05). However, the tear concentration in epidermal growth factor (EGF), fractalkine, IL-8, IL-1RA, interferon-inducible protein (IP)-10/CXCL10, vascular endothelial growth factor (VEGF) and IL-6, comparing the active and inactive period, was not statistically different. Apart from the tear VEGF levels, the cytokine/chemokine concentration in tears in the active/inactive phase was statistically different (*p* < 0.05) from the counterpart levels in plasma. In conclusion, no isolated cytokine/chemokine in the tears has been found in a concentration which could be used as a potential biomarker of disease activity and treatment response.

## 1. Introduction

The term uveitis refers to the inflammation of the uvea tissue, which is composed of the iris, ciliary body and choroid, and it frequently involves the impairment of adjacent structures in the eye such as the cornea, vitreous humour, retina, or optic nerve. It is a severe eye disease, as it has been put forward as the cause of up to 10% of legal blindness in the USA [1]. Furthermore, several recent studies have pointed out the significant impact it has on the economy and society since it significantly affects patients in working age and diminishes their vision-related quality of life [2,3,4,5,6].

The correct approach to uveitis patients is usually a challenge to the physician for several reasons. Firstly, it encompasses a great variety of pathologies, either systemic or non-systemic, which may present with a wide range of clinical manifestations, sometimes resulting in a misleading diagnosis and consequently delaying correct treatments. Secondly, a significant percentage of uveitis has a considerable risk for sight-threatening complications, some of which can be successfully treated, such as the formation of cataracts, whereas others result in permanent vision loss [7,8,9,10]. Thirdly, being a natural-chronic disease, uveitis frequently needs chronic and expensive treatments requiring close follow-up, which could burden health services [2].

Although the exact pathophysiological mechanism underlying uveitis has not been completely unveiled, the cytokine and chemokine balance appears to have a critical role in the pathway of this entity. Cytokines and chemokines are secreted proteins mainly derived from mononuclear phagocytic cells and T-lymphocytes’ growth, differentiation and activation functions. They regulate all immune and inflammatory responses, acute and chronic, being a crucial mediator in cytotoxic, humoural, cell-mediated and allergic immunity, and their imbalance could lead to a real impairment in the immune system response [11]. They can be classified by the nature of the responses they mediate, pro- or anti-inflammatory signalling, although their activity can vary depending upon cell type and location. For instance, Interleukin (IL)-1, IL-6 or tumour necrosis factor (TNF)- α are key pro-inflammatory cytokines. On the other side, IL-12 and IL-10 are considered inhibitors and IL-2 regulatory [12].

Within this framework, several studies have reported increased levels of cytokines/chemokines in either plasma, vitreous or aqueous humour (Aqh) [13,14,15,16,17,18,19]. Based on these studies, it has been proposed that different diseases may manifest with different cytokine profiles, which could aid in the diagnosis of specific cases.

In spite of the fact that the outcomes provided by the literature suggest that the analysis of cytokines/chemokines could be of utmost importance in our daily routine, it is also true that the procedures used to obtain these outcomes are not without risk, as they are invasive procedures which could be challenging to apply in the clinical setting. This is why the study of non-invasive tests that could have clinical implications is a field of interest.

Being a non-invasive approach, analysis of the cytokines/chemokines concentration in tears could improve our understanding of the ocular immunological changes that take place over the ocular surface in this entity, which could also provide the possibility of more specific and effective treatments according to the stage of the disease [20,21,22]. Prompted by this theory, we have conducted an observational prospective analytical study based on two main hypotheses: Firstly, that the cytokines/chemokines profile in tear samples would vary according to the uveitis activity. Secondly, active uveitis could also express different cytokines/chemokines profiles in tears compared with their counterpart in the cytokines/chemokines profile in plasma.

## 2. Materials and Methods

### 2.1. Patients

An observational prospective longitudinal analytical study was performed. The present study was approved by the Institute of Applied Ophthalmobiology (IOBA) institutional review board and the University of Valladolid Clinical Hospital Ethics Committee and followed the tenets of the Declaration of Helsinki. Study participants were selected among patients who attended the Ocular Immunology and Uveitis Unit of both IOBA and University of Valladolid Clinical Hospital with active uveitis between July 2018 and January 2020. Informed consent was obtained from each participant over 18 years old, and parental consent was provided in the case of patients under that age.

Clinical evaluation was always performed by authors LC and LG, each one always evaluating the same clinical parameters: slit-lamp anterior chamber cells, anterior chamber flare, vitreous haze grade, intraocular pressure, and estimation of best-corrected visual acuity in logMAR format. All patients, regardless of the type of uveitis, anatomic classification and grade of inflammation, were evaluated according to the criteria proposed by the Standardisation of Uveitis Nomenclature (SUN) Working group [23]. In addition, the difference uveitis aetiologies were recorded (Table 1). In the case of bilateral uveitis, only the most affected eye was included.

Eligible patients had a clinical diagnosis of active anterior uveitis in the first visit and started on dexamethasone 0.1% eye drops six times a day and cyclopentolate 1% three times a day until the suppression of uveitis was achieved, then topical corticosteroids were tapered and cyclopentolate 1% was stopped. Study visits occurred at baseline, and weeks 2, 6, 12 and 24. The presence of anterior chamber cells for more than 1+, a two-step increase in the vitreous haze, or newly formed retinal/choroidal lesions at least three months after the end of corticosteroid treatment were considered as an active uveitis and excluded from the study [23]. Therefore, only patients with inactive uveitis on the last clinical examination were included.

To ensure as much as possible that the results would not be influenced by the effects of topical medications and/or any other surface disease, exclusion criteria were established. Exclusion criteria were the presence of any other active ocular surface disease at the time of sample collection, such as infectious conjunctivitis, keratitis, allergy or dry eye disease, and the presence of active uveitis during the last visit. In addition, patients with intermediate or posterior uveitis were excluded from the study as well.

### 2.2. Tear Sample Collection

Single tear samples were obtained from the active-uveitis eye in both the first and the last visit. Following the technique previously described by our group [24], we used a capillary glass tube (Drummond Scientific Co., Broomall, PA, USA) to collect a 1- μL tear sample from the external canthus, avoiding tear reflex as much as possible. The collected sample was then diluted 1:10 in a sterile tube containing a cold cytokine assay buffer (Millipore Ibérica, Madrid, Spain), and stored at −80° until assayed. The samples were obtained unilaterally from the same eye in the active and inactive phase and were not pooled.

### 2.3. Serum Sample Collection

A venous blood sample was taken from the antecubital vein. The blood samples were immediately placed into individual Eppendorf tubes (Eppendorf, Fremont, CA, USA) and centrifuged at 6000 rpm for five minutes at 4 °C. The sera were stored at −80 °C until used for the assay.

### 2.4. Analysis of Tear and Serum Cytokines/Chemokines

The presence and concentration of 12 molecules were determined in both tear and serum samples by a multiplex immunobead-based array (Milliplex 12x-Human Cytokine/TH17 Magnetic Bead Panel, SPR (1504); Millipore, Watford, UK), using a Luminex IS-100 (Luminex Corporation, Austin, TX, USA). The following molecules were assayed: epidermal growth factor (EGF), fractalkine, interleukin (IL) 1 receptor antagonist (IL-1RA), IL-1 *β*, IL-17A, IL-2, IL-23, IL-6, IL-8/CXCL8, interferon-inducible protein (IP)-10/CXCL10, vascular endothelial growth factor (VEGF) and tumour necrosis factor-alpha (TNF-α).

The samples were analysed as previously described [25] following the manufacturer’s protocol, taking into consideration the required modifications for low volume assays for the case of the tear samples analysis. In summary, 10 μL of a final 1:10 diluted tear sample and 25 μL of a non-diluted plasma sample were incubated with 25 μL of anti-body-coated capture beads for 12 h at 4 °C.

The collected data were stored and analysed using the “BEAD VIEW Software” (Upstate-Millipore, Watford, UK). The minimum detectable concentrations (provided by the assay manufacturer) were (in pg/mL) EGF = 4.47; fractalkine = 4.27; IL-1Ra = 10.97; IL-1 *β* = 0.4; IL-17A = 0.25; IL-2 = 0.3; IL-23 = 28.6; IL-6 = 1.1; IL-8/CXCL8 = 0.4; (IP)-10/CXCL10 = 1.14; VEGF = 2.6; TNF-α = 1.1. For statistical analysis, in cases where the assayed molecule was undetectable, a regression-based imputation method for missing values was used, based on the multivariate imputation by chained equations algorithm [26]. Cytokine expression data were transformed using the logarithmic base 2 scale.

### 2.5. Statistical Analysis

Statistical analyses were performed by LGO using the R software (R Foundation for Statistical Computing, Vienna, Austria).

Results are expressed as mean ± SD unless otherwise indicated. Normality assumptions were checked by the Shapiro–Wilk test. Comparison between samples was performed using either Student’s paired 2-tailed *t*-tests for parametric data or the Wilconxon’s signed rank sum test, the respective nonparametric test. *p* values less than or equal to 0.05 were considered significant.

## 3. Results

### 3.1. Clinical Data

Of the 36 patients initially recruited, 18 were excluded for the following reasons: inability to collect tears (*n* = 6), active disease in the last follow-up appointment (*n* = 6), active intermediate or posterior uveitis (*n* = 4) or concomitant ocular surface condition (*n* = 2). Eighteen uveitis patients, 10 females (66.6%) and 8 males (44.4%), of age 52.5 ± 12.2 years, were included in the study.

Fifteen of the tear and plasma samples (83.3%) came from patients with unilateral uveitis, while three (16.7%) came from patients with bilateral uveitis, in which case only the eye with more severe condition was considered. Clinical data and uveitis characteristics are shown in Table 1.

Fifteen patients (83.3%) were not receiving systemic treatment at the time of sample collection, neither during the first episode nor at the time of final evaluation. Three samples (13.6%) were from patients on systemic antiviral treatment at the time of sample collection either during the first episode or at the time of final evaluation. Two (9.1%) were on a systemic anti-TNF-α agent.

### 3.2. Cytokine and Chemokine Percentages of Detection in Tear Samples

Seven out of twelve molecules assessed in the tear samples were detected in more than 60% of the samples in the active phase: EGF, fractalkine, IL-1RA, IL-6, IL-8 and IP-10. Six out of twelve molecules were detected in more than 60% of the samples in the inactive phase: EGF, fractalkine, IL-1RA, IL-8 and IP-10. IL-6 was detected in 77.8% of the tear samples in the active phase, and it was detected in 38.9% of the counterpart inactive phase samples, the difference being statistically significant (*p* < 0.05) (Figure 1).

### 3.3. Cytokine and Chemokine Percentages of Detection in Plasma Samples

Eight out of twelve molecules assessed in the plasma samples were detected in more than 60% of the samples in the active phase: EGF, fractalkine, IL-1RA, IL-6, IL-8, IP-10, TNF-α and VEGF. Eight out of twelve molecules assessed in the plasma sample were detected in more than 60% of the samples in the inactive phase: EGF, fractalkine, IL-1RA, IL-6, IL-8, IP-10, TNF-α and VEGF (Figure 2).

### 3.4. Cytokine and Chemokine Tear Levels According to Uveitis Activity

According to the uveitis activity, there were no differences in the concentration of EGF, fractalkine, IL1R**α**, IL-8, IP-10, VEGF and IL-6 in the tears, comparing the active and inactive periods (Table 2).

### 3.5. Cytokine and Chemokine Plasma Levels According to Uveitis Activity

According to the uveitis activity, there were no differences in the plasma concentration of EGF, Fractalkine, IL1-*β*, IL-17A, IL1RA, IL-2, IL-23, IL-8, IP-10, TNF-α, VEGF and IL-6, comparing the active and inactive period (Table 3).

### 3.6. Comparison of Cytokine and Chemokine Concentrations in Tears and Plasma Measured in the Active Phase

The levels of EGF, fractalkine, IL1-*β*, IL-17A, IL-1RA, IL-2, IL-23, IL-8, IP-10, TNF-α and IL-6 in patients with uveitis in their active phase were independent to their counterpart levels in plasma, the difference being statistically significant (*p* < 0.05). However, the difference between the tear and plasma levels of VEGF was not statistically significant (*p* > 0.05) (Table 4).

### 3.7. Comparison of Cytokine and Chemokine Concentration in Tears and Plasma in the Inactive Phase

The levels of EGF, fractalkine, IL1-*β*, IL-17A, IL-1RA, IL-2, IL-23, IL-8, IP-10, TNF-α and IL-6 in patients with uveitis in their active phase were independent of their counterpart levels in plasma, the difference being statistically significant (*p* < 0.05). However, the difference between the tear and plasma level of VEGF was not statistically significant (*p* > 0.05) (Table 5).

## 4. Discussion

One of the cornerstones of the most widely accepted classification system of uveitis, designed by The SUN Working group in 2005, consists of grading the severity of inflammation, which is the most important when assessing the efficacy of different treatment approaches [23]. However, the degree of inflammation is a subjective parameter, and the development of new objective techniques to determine uveitis activity is a matter of interest. In this study, we intended to identify molecules specific to active uveitis, and to do this, we analysed the tear levels of a panel of cytokines/chemokines in the tears and plasma of patients with uveitis and compared those levels in the active or inactive phase of the disease. To the best of our knowledge, there are no studies assessing the tear cytokine/chemokine levels in uveitis patients according to their uveitis activity.

Regarding the aetiology, it is vital to consider that the aetiology of the uveitis could vary the cytokine/chemokine profile in tears and plasma. Therefore, sampling patients with different aetiologies could bias our outcomes and should be taken into consideration in future investigations, as the activation of different molecular pathways could lead to anterior uveitis. For instance, Takase et al. [27] investigated the cytokine profile in the aqueous humour and sera of patients with infectious and non-infectious uveitis, suggesting that cytokines in infectious uveitis were locally produced while in non-infectious uveitis they seem to be produced in the eye and peripheral blood. However, this is not the scope of our investigation, and further studies with a high number of patients are required.

In our study, a panel of 12 inflammatory molecules was analysed in the tears and plasma of the patient throughout the different stages of the uveitis, which led us to investigate the variability of the molecules according to the uveitis activity and the relationship between the plasma and tear levels, as little is known about the origin of the cytokines in the tear. The technological advancements in the analysis of cytokines and chemokines have allowed the investigation of the cytokine profile from a very small sample such as vitreous or aqueous humour, or plasma [13,14,15,16,17,18,19,27,28,29]. Our study revealed that only seven out of twelve molecules were detected in more than 60% of tear samples, which raises an important question about which particular cytokines should be investigated in tears.

For instance, TNF-α was detected in tears in less than 10% of the samples, which suggests there might be no link between the concentration of TNF-α in tears and the intraocular levels, which is of particular interest as TNF-α inhibitors are currently one of the most common treatments in non-infectious uveitis. Hence, adalimumab, a fully human monoclonal antibody, is the only biologic agent approved by the FDA to treat non-anterior non-infectious uveitis [30]. In addition, in 2014, the American Uveitis Society supported the use of biologic agents (infliximab and adalimumab) as a first- or second-line immunomodulatory therapy, depending on the cause underlying the non-infectious uveitis, in the treatment of sight-threatening non-infectious uveitis. Our outcome is concordant with previous uveitis investigations in which the tear profile was analysed, emphasising that tear TNF-α levels do not reflect the concentrations within the eye [21,31]. However, we must consider that the TNF-α percentage of detection in tears is not constant in the literature, varying from 1% to 100% [32,33,34,35,36]. We have to consider that the TNF-α detection rate could be affected by different factors. For instance, it has been shown that there is an intra-day variation of TNF-α in healthy patients throughout the day, with increased levels in the morning and late in the day [32,33], although it is unclear if these results could be the same in uveitis patients. This should be considered as, in a real-case scenario, we cannot control the time onset of the presentation of the patient in the emergency clinic, which could bias our results. In addition, the fact that the tear samples were diluted 1/10 for analysis could explain some of our low rates of TNF-α detection [33]. However, it has not affected other cytokines and should not bias our outcomes, although it could be a matter of controversy. Furthermore, cytokine analysis in tear samples is an area of continuous evolution. New technology is constantly being introduced, with new ways of collecting, storing and analysing, which could affect our capacity to compare different studies. Therefore, we believe that further research on establishing a common approach for cytokine tear analysis in patients with uveitis should be carried out. It could help us resolve all the remaining unsolved questions [21,29].

On the other hand, the percentage of IL-6 detection showed a lower tendency of detection after finishing the treatment. In previous investigations, the aqueous humour level of IL-6 has been correlated with the number of lymphocytes and neutrophils [13,37]. The IL-6 is a pleiotropic cytokine that interferes in many biological processes such as hematopoiesis, immune defence, and neovascularization. Therefore, it has been the focus of different studies due to its role in the immune cascade, and its pathological dysregulation in the inflammatory autoimmune system and ocular disease is considered a potential biomarker and target of new therapies [38].

The development of new biomarkers has been the focus of multiple investigations as the assessment of disease severity or treatment effect could be a game-changer in managing eye conditions. Recently, studies have suggested that the different cytokines and chemokines levels in aqueous humour, vitreous and serum of uveitis patients could be a potential biomarker [13,37,39,40,41,42,43,44]. However, the potential limitations of these studies are also known, as the obtaining of aqueous and vitreous samples carried significant risk, and it remains unclear whether the expression of cytokines and chemokines in the serum is the same in the eye. Hence, new studies have explored the potential value of tear samples [20,21,22,31,45,46], as it is a non-invasive procedure; it is hypothesised that it could be used not only as a possible biomarker of uveitis activity but also as a predictive biomarker of response to the treatment, but the results have been controversial.

Within this framework, Carreño et al. [21] evaluated a panel of 21 inflammatory molecules in tears of uveitis and healthy patients, finding increased levels of IL-1RA, IL-8/CXCL8, fractalkine/CX3CL1, IP-10/CXCL10, VEGF and TGF-*β*2 in uveitis patients. However, they could not find significant differences in cytokine tear levels between active and inactive uveitis. To confirm or refute these data, we decided to evaluate the hypothesis that the tear molecules’ concentration could reflect the uveitis activity. Still, our outcomes could not find significant results in concordance with previous studies.

Türkçüoğlu et al. [45] investigated the potential association between the IL-2 tear and serum levels and the Behçet disease activity, finding no association of disease activity with serum and tear IL-2 levels. In the same way, Shirinsky et al. [31] evaluated the possible link between the IL-6, IL-8 and IL-10 cytokine tear levels and eye inflammation during treatment, and they did not find any link between the tear cytokine levels and changes in eye inflammation during the treatment. However, they reported that the baseline IL-10 concentration was a significant and independent predictor of clinical response to treatment.

Our results are in concordance with these investigations, as no significant changes have been observed in the concentration of the chemokines and cytokines in tears. However, as mentioned above, it is essential to highlight the low tendency in the percentage of detection of IL-6 in the inactive phase, being detected in two-fold more cases in the active phase. In our study, the potential role of IL-10 as a clinical predictor was not assessed as patients who showed any degree of activity during the 6-month time appointment were excluded. Therefore, further investigations trying to unveil possible predictor factors could benefit the clinical practice.

On the other hand, other studies have found different protein profiles in the tears of patients with active uveitis. For example, Eidet et al. [47] analysed the tear fluid proteome of eyes with unilateral acute anterior uveitis using their healthy eyes as control. They concluded that the concentration of molecules involved in the inflammation-associated LXR/RXR pathway might differ between uveitis eyes and their counterpart in healthy eyes. Analysing their outcomes, they showed increased levels of TNF-α and IL-6 in four out of five eyes with uveitis, which is contradictory to our study. This study may differ in findings since Eidet et al. [47] compared the protein level found in the tear fluid of the diseased eye relative to the contralateral-non-uveitis eye of patients with unilateral acute anterior uveitis, while our investigation is based on the potential changes in the tear in two different stages of the disease in the same eye.

In addition, Kumar et al. [20] analysed the cytokines tear levels in HLA-B27-related uveitis. They also concluded that not only could IL-6 be a potential biomarker associated with the entity, but it could also be used as a prognosis factor, although new research confirming their outcomes should be necessary to verify their findings. Further investigations with a higher number of patients and a correlation between the grade of activity and its chronicity could be of interest, as it has been stated that IL-6 and IL-8 tear levels are highly correlated with the duration of chronic eye inflammation [31].

Regarding the potential role of the serum cytokine profile, many investigations have explored the correlation between different uveitis clinical entities and disease activities and serum cytokine levels with contradictory results. Of note, most of these investigations have compared active with inactive uveitis or a healthy control group showing increased serum levels of different cytokines in the active group, such as the IL-8 [39,40], IL-6 [39], TNF-α [41], IL-17 [42], IL-21 [43], IL-23 [42,44]. However, few prospective investigations have unveiled the cytokine profile change according to the uveitis activity and response to the treatment [48,49]. In this framework, Cordero-Coma et al. [49] carried out a prospective study to investigate the effects of adalimumab on the serum cytokine profile in patients with steroid-treated recurrent uveitis, unveiling that reduced IL-22 levels were the only cytokine correlated with disease activity, without finding this effect on steroid-treated patients. As far as we know, our study is the first one in which a prospective evaluation of the serum cytokine profile and its correlation with uveitis activity in patients without immune-modulator treatments was attempted without unveiling any correlation between the serum–cytokine profile and the uveitis activity. Cordero-Coma et al. [49] only found a correlation in the IL-22 level with disease activity in patients treated with adalimumab, while no correlation was found in steroid-treated patients; our investigation has been carried out on patients treated with standard steroid treatment. Therefore, further studies are needed to clarify if the serum cytokine profile could be used as a biomarker in refractory cases treated with immunomodulator or biological drugs.

Of note, unveiling new non-invasive biomarkers could be a field of particular interest in patients affected by juvenile idiopathic arthritis (JIA) with eye manifestations, as it commonly manifests as chronic anterior uveitis, which is often clinically silent and can lead to serious visual impairment [50]. Angeles-Han et al. [51] showed a different proteomic tear profile between patients with JIA-associated uveitis and healthy patients, S100, IL-8 and soluble intracellular adhesion molecule (sICAM1), a potential biomarker for disease activity. Moreover, it has been shown that the proteomic profile between patients with JIA-associated uveitis and idiopathic uveitis could also be different [22,52]. Therefore, further studies unveiling the potential role of the tear proteomic profile in paediatric patients could be game-changing in their management [47].

The cytokine and chemokine tear origin is still controversial, as there is no clear answer as to whether the cytokines in tears are derived from the lacrimal gland, secreted by the conjunctival epithelium, by leakage from the surrounding blood vessels [53], or in a mixed manner. These unresolved questions led us to investigate whether the cytokine and chemokine tear profile expressed was similar or not to the cytokine and chemokine plasma profile. This study has shown that, apart from the VEGF concentration, the concentration of cytokines and chemokines in tear samples was independent of their counterpart in serum samples. These results might suggest that the VEGF plasma concentration affects the tear levels, which is in concordance with previous studies in diabetic patients [54]. On the other side, the tear concentration of EGF, fractalkine, IL-1RA, IL-8, IP-10 and IL-6 has been shown to be higher compared with the plasma profile in both active and inactive uveitis, which could be related to a local dysregulation of the immune system in the eye, leading to an increased local production of these cytokines, although further investigations are needed to unveil the origin of each cytokine in the tears.

In summary, to our knowledge, this is the first study in uveitis investigating whether the cytokine and chemokine concentration in tears could be a potential biomarker for uveitis activity or treatment response without showing significant changes between the active and inactive phases in the molecules studied. In addition, we have tried to unveil whether or not the cytokine and chemokine profiles in the tear samples are related to their concentration in plasma or if they were independent, as there is some controversy about the cytokine and chemokine tear origin, being postulated that it is a reflection of the one in the plasma. However, we have not found any correlation in these levels apart from the VEGF, the only molecule whose tear concentration showed a significant relationship with the plasma concentration. The discrepancy between the TNF-α levels in tear and plasma samples is remarkable since, as mentioned above, one of the main treatments is non-infectious uveitis based on blocking this molecule.

The main limitations of this study are the low number of patients included in the study, not enough to permit a proper subanalysis according to the type of uveitis, and the restricted number of molecules studied per sample, limiting a more completed analysis of the tear profile. In addition, we have to consider that uveitis is a rare condition with an estimated prevalence of around 121 cases per 100,000 adults, responsible for 10% of cases of blindness in the United States [55]. Moreover, we have analysed a heterogenous sample and cannot generalise our results as a different cytokine concentration has been found based on uveitis localisation, activity and presence [21]. We have to consider that our investigation has evaluated only anterior uveitis. Hence, these results cannot be extrapolated to all uveitis, as different localizations could lead to different cytokine profiles.

One of the limitations of studying small samples is the limited number of molecules evaluated per sample. In this framework, the Luminex method has been used to investigate the potential benefit of the cytokine and chemokine profiles in tears due to its capacity to assess several proteins with one sample. However, new methods such as the quantitative microarray or the TripleTOF 5600 system allow a greater range of molecules to be evaluated per sample, which could be of interest for a more general analysis of the tear profile in uveitis patients according to their degree of activity [56,57].

Considering everything, our conclusions are tentative, as multiple conditions could interfere with the outcomes. Therefore, further studies are needed to unveil the potential benefit of analysing the tear proteome to monitor treatment response or uveitis activity.

## Figures and Tables

**Figure 1 jcm-11-07034-f001:**
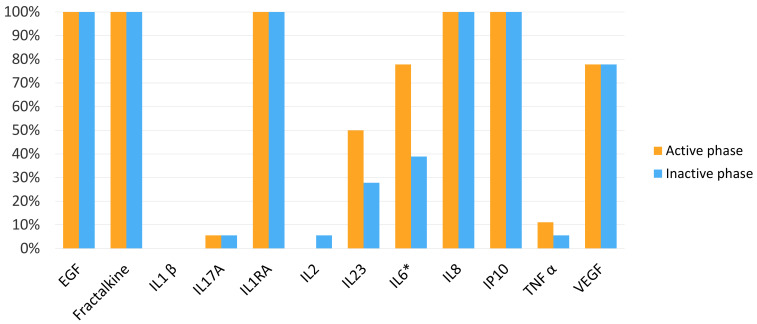
Percentage of epidermal growth factor (EGF), fractalkine, interleukin (IL) 1 receptor antagonist (IL-1RA), IL-1 *β*, IL-17A, IL-2, IL-23, IL-6, IL-8/CXCL8, interferon-inducible protein (IP)-10/CXCL10, vascular endothelial growth factor (VEGF) and tumour necrosis factor-alpha (TNF-α) detection in tears in the active phase and inactive phase. * The difference was considered statistically significant when *p* < 0.05.

**Figure 2 jcm-11-07034-f002:**
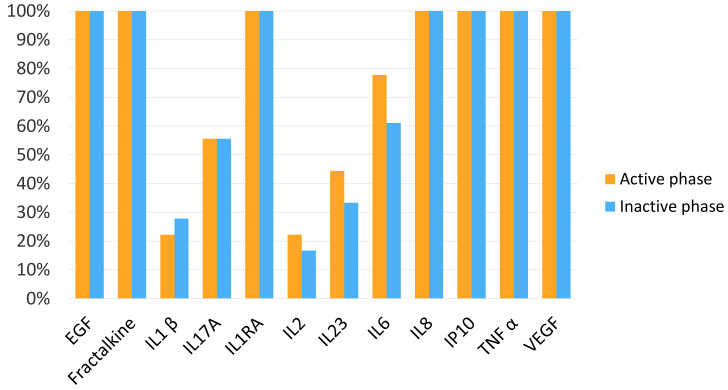
Percentage of epidermal growth factor (EGF), fractalkine, interleukin (IL) 1 receptor antagonist (IL-1RA), IL-1 *β*, IL-17A, IL-2, IL-23, IL-6, IL-8/CXCL8, interferon-inducible protein (IP)-10/CXCL10, vascular endothelial growth factor (VEGF) and tumour necrosis factor-alpha (TNF-α) detection in plasma in the active phase and inactive phase.

**Table 1 jcm-11-07034-t001:** Clinical data of uveitis patients.

**Age mean (± SD)**	52.5 (±12.2)
**Gender** * **n** * **(%)**	
Male	8 (44.4)
Female	10 (66.6)
**Anatomical location of uveitis** * **n** * **(%)**	
Anterior	18 (100)
**Distribution** * **n** * **(%)**	
Unilateral	15 (83.3)
Bilateral	3(16.7)
**Underlying diagnosis** * **n** * **(%)**	
Ankylosing spondylitis	3 (16.7)
Herpetic uveitis	3 (16.7)
Psoriatic arthritis	1 (5.5)
Fuchs Heterochromic Iridocyclitis	1 (5.5)
Idiopathic	10 (55.5)
**Total** * **n** * **(%)**	18 (100)

Clinical data and distribution of uveitis according to anatomic classification, distribution and aetiology. SD: Standard deviation.

**Table 2 jcm-11-07034-t002:** Cytokine and chemokine tear levels according to uveitic activity.

	Active Uveitis Mean ± SD (pg/mL)	Inactive Uveitis Mean ± SD (pg/mL)	*p* Value
**EGF**	563.09 ± 22.70	625.90 ± 26.41	0.789
**Fractalkine**	1633.50 ± 145.51	912.69 ± 150.12	0.181
**IL-1RA**	5592 ± 120.50	4330.90 ± 134.80	0.653
**IL8**	148.41 ± 34.90	187.40 ± 40.50	0.640
**IP10**	8745.45 ± 100.98	10,102.50 ± 150.76	0.655
**VEGF**	97.12 ± 9.90	98.12 ± 8.98	0.316
**IL-6**	34.68 ± 3.76	34.88 ± 1.70	0.396 ^†^

Concentration of epidermal growth factor (EGF), fractalkine, interleukin (IL) 1 receptor antagonist (IL-1RA), IL-8/CXCL8, IP-10/CXCL10, vascular endothelial growth factor (VEGF) and IL-6 in tears in the active phase and inactive phase. The difference was considered statistically significant when *p* < 0.05. SD: Standard deviation, ^†^ Wilconxon’s test.

**Table 3 jcm-11-07034-t003:** Cytokine and chemokine plasma levels according to uveitic activity.

	Active Uveitis Mean ± SD (pg/mL)	Inactive Uveitis Mean ± SD (pg/mL)	*p* Value
**EGF**	167.93 ± 35.64	207.98 ± 25.68	0.572
**Fractalkine**	477.39 ± 46.51	426.57 ± 52.59	0.122 ^†^
**IL1-** * **β** *	2.02 ± 1.49	3.02 ± 1.92	0.335 ^†^
**IL-17A**	7.6 ± 4.35	8.02 ± 2.99	0.813 ^†^
**IL-1RA**	26.10 ± 12.43	29.28 ± 10.31	0.116 ^†^
**IL-2**	2.09 ± 1.53	2.06 ± 1.94	0.260 ^†^
**IL-8**	15.38 ± 10.41	16.30 ± 12.49	0.922
**IP-10**	306.78 ± 102.01	268.32 ± 90.95	0.952
**TNF-α**	7.94 ± 4.60	8.94 ± 3.70	0.842
**VEGF**	159,75 ± 30.89	169.38 ± 40.70	0.849 ^†^
**IL-6**	13.06 ± 6.03	207.98 ± 25.68	0.731 ^†^

Concentration of epidermal growth factor (EGF), fractalkine, interleukin (IL) IL1-*β,* IL-17A, interleukin (IL) 1 receptor antagonist (IL-1RA), IL-2, IL-23 IL-8/CXCL8, IP-10/CXCL10, tumour necrosis factor-alpha (TNF-α), vascular endothelial growth factor (VEGF) and IL-6 in plasma in the active phase and inactive phase. The difference was considered statistically significant when *p* < 0.05. SD: Standard deviation, ^†^ Wilconxon’s test.

**Table 4 jcm-11-07034-t004:** Comparison of cytokine concentrations in tears and plasma in the active phase.

	Tear Sample Mean ± SD (pg/mL)	Plasma Sample Mean ± SD (pg/mL)	*p* Value
**EGF**	563.09 ± 22.70	167.93 ± 35.64	**0.000**
**Fractalkine**	1633.50 ± 145.51	477.39 ± 46.51	**0.000** ^†^
**IL1-** * **β** *	-	2.02 ± 1.49	-
**IL-17α**	-	7.6 ± 4.35	-
**IL-1RA**	5592.02 ± 120.50	26.10 ± 12.43	**0.000**
**IL-2**	-	2.09 ± 1.53	-
**IL8**	148.41 ± 34.90	15.38 ± 10.41	**0.000**
**IP10**	8745.45 ± 100.98	306.78 ± 102.01	**0.000**
**TNF-α**	-	7.94 ± 4.60	-
**VEGF**	97.12 ± 9.90	159,75 ± 30.89	0.815 ^†^
**IL-6**	34.68 ± 3.76	13.06 ± 6.03	**0.000** ^†^

Concentration of epidermal growth factor (EGF), fractalkine, interleukin (IL) IL1-*β,* IL-17A, interleukin (IL) 1 receptor antagonist (IL-1RA), IL-2, IL-23 IL-8/CXCL8, IP-10/CXCL10, tumour necrosis factor-alpha (TNF-α), vascular endothelial growth factor (VEGF)and IL-6 in tears in the active phase compared to their counterpart levels in plasma in the active phase. The difference was considered statistically significant when *p* < 0.05. SD: Standard deviation, ^†^ Wilconxon’s test. Bold numbers are the ones showing a statistical significant difference.

**Table 5 jcm-11-07034-t005:** Comparison of cytokine concentrations in tears and plasma in the inactive phase.

	Tear Sample Mean ± SD (pg/mL)	Plasma Sample Mean ± SD (pg/mL)	*p* Value
**EGF**	625.90 ± 26.41	207.98 ± 25.68	**0.000**
**Fractalkine**	912.69 ± 150.12	426.57 ± 52.59	**0.000**
**IL1-** * **β** *	-	3.02 ± 1.92	-
**IL-17A**	-	8.02 ± 2.99	-
**IL-1RA**	4330.90 ± 134.80	29.28 ± 10.31	**0.000** ^†^
**IL-2**	-	2.06 ± 1.94	-
**IL-8**	187,40 ± 40.50	16.30 ± 12.49	**0.000**
**IP-10**	10,102.50 ± 150.76	268.32 ± 90.95	**0.000**
**TNF-α**	-	8.94 ± 3.70	-
**VEGF**	98.12 ± 8.98	169.38 ± 40.70	0.212 ^†^
**IL-6**	34.88 ± 1.70	12.94 ± 5.18	**0.000** ^†^

Concentration of epidermal growth factor (EGF), fractalkine, interleukin (IL) IL1-*β,* IL-17A, interleukin (IL) 1 receptor antagonist (IL-1RA), IL-2, IL-23 IL-8/CXCL8, IP-10/CXCL10, tumour necrosis factor-alpha (TNF-α), vascular endothelial growth factor (VEGF)and IL-6 in tears in the inactive phase compared to their counterpart levels in plasma in the inactive phase. The difference was considered statistically significant when *p* < 0.05. SD: Standard deviation, ^†^ Wilconxon’s test. Bold numbers are the ones showing a statistical significant difference.

## Data Availability

Not applicable.

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
