# Peer review of "Tear and Plasma Levels of Cytokines in Patients with Uveitis: Search for Active Disease Biomarkers"

_jcm, 2022, doi:10.3390/jcm11237034_

Round 1

Reviewer 1 Report

1) It is necessary to provide some specific comments, such as: What isthe main question addressed by the research? How original is the topic?What does the study add to the subject area compared with otherpublished material? Are the conclusions consistent with the evidence andarguments presented? Do the conclusions address the main question posed?

This research paper evaluates the utility of tear cytokine levels as a non-invasively obtained biomarker for uveitis activity.

This subject has been evaluated by prior studies, including Carreno et al. and Shirinsky et al, that are cited in this paper. While Carreno et al found a difference in tear cytokine levels between patients uveitis and those without, there has not been an obvious correlation found between various tear cytokine levels and uveitic activity in prior studies.

This paper attempts to expand on prior studies by also comparing tear cytokine levels to serum cytokine levels. However the results are presented in a somewhat confusing manner and the clinical relevance is not clear. Most importantly, serum cytokine levels have not been shown to reliably correlate with uveitic activity, so the usefulness of even doing a comparison between tear and serum cytokine levels as an attempt to indirectly measure uveitic activity does not seem high. Tables 4 and 5 show large differences in concentration levels between tear and serum cytokine levels (with concentration in the tears being generally higher than in serum) but the relevance of this is not really discussed or made clear. 

If the serum cytokine level analysis is taken out, then this paper does not seem to add much to what is already in the literature.

Author Response

Reviewer´s 1

It is necessary to provide some specific comments, such as:

Thank you for your kind comment, and let us know that some essential parts of the investigation are not completely clear. Please you could find below the lines in which these specific basics of the paper are addressed:

What is the main question addressed by the research?

The hypothesis is highlighted at the end of the introduction (Lines 73-78)

“…specific and effective treatments according to the stage of the disease20–22. Prompted by this theory, we have conducted an observational prospective analytical study based on two main hypotheses: Firstly, the cytokines/chemokines profile in tear samples would vary according to the uveitis activity. Secondly, active uveitis could also express different cytokines/chemokines profiles in tears compared with their counterpart in plasma the cytokines/chemokines profile.”

How original is the topic?
What does the study add to the subject area compared with other published material?

From our point of view, finding non-invasive biomarkers for uveitis is a current topic, as these new biomarkers could help develop and testing new drugs, helping us in the management and treatment of the patient. In addition, as far as we know, this is the first longitudinal prospective study in which both tear and serum from the patients are investigated. 

Are the conclusions consistent with the evidence and arguments presented? Do the conclusions address the main question posed?

At the end of the discussion, we tried to answer the main question, which is if the active uveitis could express different cytokine cytokines/chemokines profiles in tears compared with their counterpart in plasma the cytokines/chemokines profile. We conclude that this research does not show such a difference. However, new technology is in development which could improve how accurate we are and help us to determine if the cytokine/chemokine profile could be of interest. (Line 416-442)

 In summary, to our knowledge, this is the first study in uveitis investigating whether the cytokine and chemokine concentration in tears could be a potential biomarker for uveitis activity or treatment response without showing significant changes between the active and inactive phases in the molecules studied. In addition, we have tried to unveil if the cytokine and chemokine profile in the tear samples is related to their concentration in plasma or if they were independent, as there is some controversy about the cytokine and chemokine tear origin, being postulated that it is a reflection of the one in the plasma. However, we have not found any correlation in these levels apart from the VEGF, the only molecule whose tear concentration showed a significant relation with the plasma concentration. The discrepancy between the TNFα levels in tear and plasma samples is remarkable since, as mentioned above, one of the main treatments is non-infectious uveitis based on blocking this molecule.

The main limitations of this study are the low number of patients included in the study, not enough to permit a proper subanalysis according to the type of uveitis, and the restricted number of molecules studied per sample, limiting a more completed analysis of the tear profile. In addition, we have to consider that uveitis is a rare condition with an estimated prevalence of around 121 cases per 100 000 adults, responsible for 10% of cases of blindness in the United States 56. Moreover, we have analysed a heterogenous sample and cannot generalise our results as a different cytokine concentration has been found based on uveitis localisation, activity and presence 21.

One of the limitations of studying small samples is the limited number of molecules evaluated per sample. In this framework, the Luminex method has been used to investigate the potential benefit of the cytokine and chemokine profile in tears due to its capacity to assess several proteins with one sample. However, new methods such as the quantitive microarray or the TripleTOF 5600 system allow a greater range of molecules to be evaluated per sample, which could be of interest for a more general analysis of the tear profile in uveitis patients according to their degree of activity 57,58.

This research paper evaluates the utility of tear cytokine levels as a non-invasively obtained biomarker for uveitis activity.

This subject has been evaluated by prior studies, including Carreno et al. and Shirinsky et al, that are cited in this paper. While Carreno et al found a difference in tear cytokine levels between patients uveitis and those without, there has not been an obvious correlation found between various tear cytokine levels and uveitic activity in prior studies.

Thank you for mentioning one of the papers done by our group. However, this research differs from ours in the fact that this study was a transversal study and not a prospective longitudinal study, as ours. In addition, as suggested by other reviewers, we have considered only anterior uveitis, compared with Carreño et al who did not discriminate between anterior, posterior or intermediate uveitis.

This paper attempts to expand on prior studies by also comparing tear cytokine levels to serum cytokine levels. However the results are presented in a somewhat confusing manner and the clinical relevance is not clear. Most importantly, serum cytokine levels have not been shown to reliably correlate with uveitic activity, so the usefulness of even doing a comparison between tear and serum cytokine levels as an attempt to indirectly measure uveitic activity does not seem high. Tables 4 and 5 show large differences in concentration levels between tear and serum cytokine levels (with concentration in the tears being generally higher than in serum) but the relevance of this is not really discussed or made clear. 

Thank you for your comment. According to our analysis, there is no relation between cytokine tear concentration and uveitis activity. As far as we know, there is no other longitudinal study in which the cytokine tear and plasma profile have been compared in the same group of patients (longitudinal study, prospectively). These negative outcomes could help us to exclude the tear cytokine analysis as a non-invasive test for general anterior uveitis. It is possible that a subanalysis, taking into consideration the etiologies under the uveítis and with a higher number of patients, could unveil other results.

In addition, one of the unresolved questions regarding cytokine tear profile is where the cytokine concentration comes from. Our analysis tried to find out if the tear concentration profile was similar to the plasma according to the uveitis activity. However, we have not found such a similarity except in the VEGF molecule. This could mean that the origin of each molecule in the tears is different, although is a matter of controversy. A higher concentration in the tear could be related to increased local production of cytokines due to the uveitis, but it is a question that we can only hypothesize with the current knowledge. We have added our suspicion at the end of the discussion, remarking that further investigations are needed. (Line: 409-416)

“These results might suggest that the VEGF plasma concentration affects the tear levels, which is in concordance with previous studies in diabetic patients 55. On the other side, the tear concentration of EGF, fractalkine, IL-1RA, IL-8, IP-10 and IL-6 has been shown to be higher compared with the plasma profile in both active and inactive uveitis, which could be related to a local dysregulation of the immune system in the eye, leading to an increased local production of these cytokines, although further investigations are needed to unveil the origin of each cytokine in the tears.”

If the serum cytokine level analysis is taken out, then this paper does not seem to add much to what is already in the literature.

Thank you for your comment. We agree that it has been done previously but there are a couple of substantial differences that we think could add information regarding this interesting field. As mentioned above, this is the first longitudinal in which such an amount of proteins have been investigated. Compared to Shirinsky et al they just analyzed 4 cytokines, while we have analysed up to 12 cytokines. In addition, as you mentioned in your comment the serum cytokine level analysis is adding information to the literature, letting us compare different investigation groups' outcomes.

Reviewer 2 Report

I think it is very interesting and can be a starting point for future works. I would ask the authors to restrict the sample to the anterior uveitis only, as they represent the clear majority (almost 82%) in the study, so it is not possible to make evaluations regarding uveitis in general. As the authors rightly underline in the discussion, sampling patients with intermediate and posterior uveitis could bias the investigation outcomes.

Author Response

Reviewer´s point by point

Reviewer´s 2

I think it is very interesting and can be a starting point for future works. I would ask the authors to restrict the sample to the anterior uveitis only, as they represent the clear majority (almost 82%) in the study, so it is not possible to make evaluations regarding uveitis in general. As the authors rightly underline in the discussion, sampling patients with intermediate and posterior uveitis could bias the investigation outcomes.

Thank you for your kind comments. Following your recommendation, we have decided to restrict the sample to the anterior uveitis only. In addition, we have highlighted that it is possible that a subanalysis taking into consideration the natural aetiology of the uveitis should be considered in future analysis. Unfortunately, due to our restricted sample size, this subanalysis is not possible to be performed.

Several parts of the text have changed pointing out that only anterior uveitis was used in the analysis, such as:

Line 15-17: It has been hypothesised that tear levels of cytokines and chemokines could be used as a potential biomarker in patients with anterior uveitis, and it could be correlated with their concentration in plasma.”

Line 100-103: “Eligible patients had a clinical diagnosis of active anterior uveitis in the first visit and started on dexamethasone 0,1% eye drops 6 times a day and cyclopentolate 1% three times a day until the suppression of uveitis was achieved, then topical corticosteroids were tapered and cyclopentolate 1% was stopped.”

Furthermore, tables and statistical analysis have been changed accordingly

In addition, we have highlighted in the limitations that our results cannot be extrapolated to uveitis in general, as we have only taken into consideration anterior uveitis.

Line 435-437: We have to consider that our investigation has evaluated only anterior uveitis. Hence, these results cannot be extrapolated to all the uveitis, as different localizations could lead to different cytokine profiles.

Reviewer 3 Report

The study is interesting, but raises several questions that need to be clarified.

Minor issues:

- It is important to indicate the design of the study.

- The title of a table should be just that, the title. The authors include an explanation of the results shown in the table and the statistical analysis performed. This is not correct and should not be included.

- However, the authors have not included a table-foot. And this is important to explain the abbreviations used in the table, as well as the significance level that was set in the statistical analysis.

- In the methodology section in which the procedure for cytokine analysis is explained, the authors refer to a previous publication (Cocho L, 2016). However in that publication the procedure is not explained, but is refered to another previous publication (Enríquez-de-Salamanca A, 2010).

Major issue:

- The main problem is the explanation of the multiplex analysis performed.

On the one hand, the authors comment that 1µL of tear was obtained from each patient. On the other hand, authors explain that tear samples were diluted 1:10. This is 1µL tear + 9µL Assay Buffer, which makes a final volume of 10µL. But in the manufacturer's protocol it is indicated that 25µL of sample must be used. How did you manage to reach this volume? Adding more Assay Buffer to samples? In that case, the dilution used was not 1:10, but 1:25. Perhaps authors adjusted the volumes of all reagents used in the determination of cytokines. If this is what you did, how was that adjustment of volumes done?

Author Response

The study is interesting, but raises several questions that need to be clarified.

Thank you for your kind comments. Following your recommendation, we have tried to address all your suggestions.

Minor issues:

- It is important to indicate the design of the study.

Agree. We have made it clear in the methodology, by adding this sentence (Line 84):

   “An observational prospective longitudinal analytical study was performed”

- The title of a table should be just that, the title. The authors include an explanation of the results shown in the table and the statistical analysis performed. This is not correct and should not be included.

- However, the authors have not included a table-foot. And this is important to explain the abbreviations used in the table, as well as the significance level that was set in the statistical analysis.

Thank you for your comments. We have made the changes to the tables according to your suggestions. Adding the abbreviations used in the tables, and the significance level that was set in the statistical analysis.

- In the methodology section in which the procedure for cytokine analysis is explained, the authors refer to a previous publication (Cocho L, 2016). However in that publication the procedure is not explained, but is refered to another previous publication (Enríquez-de-Salamanca A, 2010).

 Thank you, we have now corrected this with the correct reference in which the procedure was explained (Line 527-529):

“25. Pinto-Fraga J, Enríquez-de-Salamanca A, Calonge M, et al. Severity, therapeutic, and activity tear biomarkers in dry eye disease: An analysis from a phase III clinical trial. Ocul Surf. 2018;16(3):368-376. doi:10.1016/j.jtos.2018.05.001”

Major issue:

- The main problem is the explanation of the multiplex analysis performed.

On the one hand, the authors comment that 1µL of tear was obtained from each patient. On the other hand, authors explain that tear samples were diluted 1:10. This is 1µL tear + 9µL Assay Buffer, which makes a final volume of 10µL. But in the manufacturer's protocol it is indicated that 25µL of sample must be used. How did you manage to reach this volume? Adding more Assay Buffer to samples? In that case, the dilution used was not 1:10, but 1:25. Perhaps authors adjusted the volumes of all reagents used in the determination of cytokines. If this is what you did, how was that adjustment of volumes done?

This reviewer is right to point out that the regular manufacturer’s protocol for multiplex cytokine assay indicates a 25 ul sample to be used. However (as already explained in the manuscript, page 3, lines 138-142), for tear cytokine analysis we have employed a manufacturer´s protocol for low-volume assays, in which only 10 ul of the sample is needed for the assay. 

We are including here the manufacturer's low-volume assay protocol for your information. 

We have now rewritten this part, hopefully, it would read now better.

“The samples were analysed as previously described (Pinto Fraga, 2018) following the manufacturer’s protocol, taking into consideration the required modifications for low-volume assays for the case of the tear samples analysis. In summary, 10 μl of a final 1:10 diluted tear sample and 25 μl of a non-diluted plasma sample were incubated with 25 ul of anti-body-coated capture beads for 12 hr at 4º C”

Round 2

Reviewer 1 Report

The authors made edits and clarifications in response to my previous concerns. The longitudinal nature of the research does differentiate it from previous similar studies.

Reviewer 3 Report

The authors have adequately answered all the questions and have considerably improved the quality of the manuscript.